# Lipidomic and Transcriptomic Reveals Variations in Lipid Deposition During Goose Fatty Liver Formation

**DOI:** 10.3390/biology14111617

**Published:** 2025-11-18

**Authors:** Qi Zhang, Chuning Bai, Mingai Zhang, Bin Yue, Jing Zhang, Min Kong, Binghan Wang, Baowei Wang, Wenlei Fan

**Affiliations:** 1College of Animal Science and Technology, Qingdao Agricultural University, Qingdao 266109, China; 2Key Laboratory of Animal (Poultry) Genetics Breeding and Reproduction, Ministry of Agriculture and Rural Affairs, State Key Laboratory of Animal Biotech Breeding, Institute of Animal Science, Chinese Academy of Agricultural Sciences, Beijing 100193, China; 3Institute of Quality Waterfowl, Qingdao Agricultural University, Qingdao 266109, China; 4Qingdao Huihe Biotechnology Co., Ltd., Qingdao 266109, China; 5College of Food Science and Engineering, Qingdao Agricultural University, Qingdao 266109, China

**Keywords:** goose fatty liver, Landes geese, overfeeding, lipidomic, transcriptomic

## Abstract

Goose foie gras is produced by feeding geese a high-energy diet, which causes their livers to become fatty. However, the exact biological process of how this fat builds up in the liver is not fully understood. This study investigated the formation of this fatty liver by analyzing geese livers before, during, and after the overfeeding period. We found that overfeeding made the livers larger, turned them yellow, and led to a massive accumulation of fat droplets inside the liver cells. By combining transcriptomic and lipidomic analyses, we identified key genes and lipid metabolites that work together to drive this process. Our study discovered four main biological pathways inside the liver cells that are responsible for the fat accumulation. These findings help explain the molecular mechanism behind goose fatty liver formation, providing crucial insight that could be used to improve breeding strategies or production efficiency for the waterfowl industry.

## 1. Introduction

Goose foie gras is a highly nutritious food, regarded as one of the world’s top three delicacies alongside caviar and black truffle [1]. Its formation relies on the force-feeding of geese with high-energy feed (primarily corn). This process disrupts the dynamic balance of lipid synthesis and leads to excessive lipid deposition within hepatocytes [2,3]. Notably, the Landes geese exhibit exceptional tolerance to hepatic steatosis, achieving fatty liver weights reaching 8–10 times greater than a normal liver [4,5].

Significant progress has been made in understanding the mechanism underlying goose fatty liver formation. Current evidence indicates this process involves a complex cascade of biological events, including activation of de novo lipogenesis pathways, restricted VLDL secretion capacity, insulin resistance, oxidative stress, and adaptive hepatocyte growth and proliferation [6,7]. Genomic studies reveal that the absence of the leptin gene in geese genome may contribute to this remarkable fat-storing capacity, potentially an evolutionary adaptation for energy reserves during long-distance migration [8]. Furthermore, in contrast to human fatty liver disease where the complement system is activated, its activity is suppressed in overfed geese [3,9], and crucially, without progression to cirrhosis or necrosis [10,11]. While these findings enhance our understanding, the precise regulatory mechanisms require further research.

The value of foie gras research extends beyond gastronomy, increasingly highlighting its potential in human health and emerging biotechnologies. Previous studies indicate that serum enzyme levels in overfed geese resemble those in human non-alcoholic fatty liver disease (NAFLD), suggesting overfed geese could serve as an alternative animal model for NAFLD research [12,13,14]. Organoid models based on goose primary hepatocytes, by simulating fructose/insulin-induced steatosis, provide novel platforms for investigating NAFLD pathology and therapeutic strategies. Moreover, the biological mechanisms enabling geese to efficiently store lipids offer critical insights for the cell-cultured meat industry. Recently, significant breakthroughs in hepatocyte culture technology have enabled the cultivation of metabolically functional adult human liver organoids from cryopreserved hepatocytes [15,16]. Applying similar organoid culture methods to geese presents a promising alternative pathway to traditional force-feeding practices.

Integration of multi-omics technologies is driving deeper mechanistic exploration of foie gras formation [17,18,19,20,21]. Transcriptomics has preliminarily outlined the temporal expression patterns of core lipid metabolism genes during the force-feeding process [22], while lipidomics has revealed the role of triglyceride accumulation in enhancing lipid droplet stability [3]. Nevertheless, the dynamic lipid molecular transformation pathways and their underlying transcriptional regulatory networks remain elusive. This study aims to integrate lipidomic and transcriptomic analysis to comprehensively investigate the dynamic changes in hepatic lipids and the underlying transcriptional regulatory mechanisms in Landes geese during different overfeeding stages. This integrated approach seeks to uncover core molecular mechanisms governing foie gras development, thereby identifying potential novel therapeutic targets for human NAFLD and establishing a molecular foundation for advancing efficient lipid engineering.

## 2. Materials and Methods

### 2.1. Animals and Sample Collection

Thirty healthy 70-day-old Landes geese were obtained from the farm of Shandong Chunguan Food Co., Ltd. (Weifang, China). The geese were housed in standardized cages (3 geese/cage, ≥0.25 m^2^/goose) with continuous water access, a space allowance consistent with established protocols in goose research [3,23]. Following a certified protocol [24], geese were force-fed a high-energy diet composed of 98% boiled corn, 1% soybean oil, 0.5% salt, and vitamins and calcium phosphate (the complete feed composition is detailed in Appendix A). The force-feeding regimen, detailed in Appendix A, began at a low frequency (once daily) and was progressively intensified to a maximum of five times per day, with corresponding increases in daily feed quantity. Samples were collected at pre-force-feeding (D0, 0 days post-initiation, n = 6), mid-force-feeding (D16, 16 days post-initiation, n = 6), and terminal-force-feeding (D25, 25 days post-initiation, n = 6). After a 6-h fast, geese were euthanized, and livers were excised and weighed. The liver index was calculated as follows: Liver index (%) = (Liver weight/Body weight) × 100%. Tissue samples were collected from the top of the liver lobes and divided into two portions. Tissue samples were collected for analysis: portions for transcriptomics, lipidomics, and chemical composition were snap-frozen, while portions for histology were fixed in 4% paraformaldehyde. All six samples per group were used for histology and composition analysis, with three randomly selected frozen samples per group used for transcriptomic and lipidomic sequencing. The study was approved by the Animal Ethics Committee of Qingdao Agricultural University (No. DKY2023001).

### 2.2. Liver Histology Analysis

Moisture content was determined using the freeze-drying method, while crude ash (CA) content was assessed through the incineration method. Crude protein (CP) content was measured by the Kjeldahl method. Crude fat (CF) content was analyzed using the Soxhlet extraction method.

### 2.3. Liver Chemical Composition Determination

Liver tissue fixed in 4% paraformaldehyde was trimmed into 1 × 1 × 1 cm^3^ blocks, dehydrated through a gradient of alcohol, and embedded in paraffin. Sections (5 μm) were prepared along the cross-sectional direction of the fibers. Staining was performed using hematoxylin and eosin, followed by sealing and preservation with neutral resin glue. The prepared HE-stained paraffin sections were examined under a light microscope, and photographs were captured [25].

### 2.4. Transcriptomic Analysis

Total RNA was isolated from liver samples using TRIzol reagent (Thermo Fisher, Wilmington, DE, USA) [26]. RNA purity, concentration and integrity was measured with a NanoDrop 2000 spectrophotometer (NanoDrop, Wilmington, DE, USA), and an Agilent 2100 Bioanalyzer (Agilent Technologies, Santa Clara, CA, USA) [27]. Following fragmentation was performed, first-strand cDNA was synthesized using random hexamers. Second strand synthesis was performed with a reaction mixture containing buffer, dNTPs, RNase H, and DNA polymerase I. The double-stranded cDNA was purified with the QIAQuick PCR Kit (QIAGEN, Hilden, Germany), followed by end repair, A-tailing, adapter ligation, and size selection. via agarose gel electrophoresis [28]. Libraries were amplified by PCR, quantified using a Qubit 3.0 Fluorometer (Thermo Fisher Scientific, Waltham, MA, USA), and assessed for insert size on an Agilent 2100 Bioanalyzer (Agilent Technologies, Santa Clara, CA, USA). Qualified libraries were sequenced on an Illumina platform PE150 (Illumina Inc., San Diego, CA, USA) [29].

### 2.5. Screening of Differentially Expressed Genes and Functional Enrichment Analysis

Initial raw read processing included quality assessment with FastQC v0.10.1 [30] and removal of adapter sequences along with low-quality bases using Trimmomatic v0.39 [31]. The HISAT2 v2.1.0 aligner [32] mapped the processed reads to the Taihu goose reference genome (https://www.ncbi.nlm.nih.gov/datasets/genome/GCF_040182565.1/; accessed on 21 June 2024). Based on the alignment results, gene expression quantification (Appendix A) was performed using RSEM v1.3.0 [33,34]. Differential expression analysis was conducted using DESeq2 v1.4.0 [35], and differentially expressed genes (DEGs) were identified with thresholds of adjusted *p*-value (*p*_adj_) < 0.05 and |log_2_(fold change)| ≥ 1. Kyoto Encyclopedia of Genes and Genomes (KEGG) pathway analyses [36,37] were performed on DEGs, with pathways at *p*-value < 0.05 considered significantly enriched.

### 2.6. Lipidomic Analysis

Total lipids were extracted from the liver samples using the methyl tert-butyl ether (MTBE) method [38]. Briefly, tissues (20 ± 1 mg) were homogenized in 1 mL extraction solvent (MTBE: MeOH = 3:1, *v*/*v*) containing internal standard. The mixture was vortexed, and after adding 200 μL of water, it was centrifuged (12,000× *g*, 10 min, 4 °C). The upper organic layer was collected, dried in a vacuum concentrator, and reconstituted in 200 μL mobile phase B for Liquid Chromatography–Tandem Mass Spectrometry (LC-MS/MS) analysis.

Lipid profiling was performed using a Liquid Chromatography–Electrospray Ionization–Tandem Mass Spectrometry system (LC-ESI-MS/MS, UPLC: ExionLC AD^TM^, Sciex, Framingham, MA, USA; MS: QTRAP^®^ 6500+ System, Sciex, Framingham, MA, USA). Chromatographic separation used a Thermo Accucore^TM^ C30 (2.6 μm, 2.1 mm × 100 mm) at 45 °C. The mobile phase consisted of (A) acetonitrile/water and (B) acetonitrile/isopropanol, both with 0.1% formic acid and 10 mmol/L ammonium formate. A 20-min linear gradient was applied at a flow rate of 0.35 mL/min, with an injection volume of 2 μL.

Mass spectrometry detection operated in both positive and negative ion modes with an ESI Turbo Ion-Spray interface. Key ESI parameters were as follows: ion source temperature, 500 °C; ion spray voltage, ±5500 V; gas pressures: 45 psi (Gas 1), 55 psi (Gas 2), and 35 psi (Curtain gas). The instrument was tuned and calibrated with polypropylene glycol solutions. Data acquisition in QQQ mode used scheduled multiple reaction monitoring (MRM), with transitions optimized using analyte-specific standards.

### 2.7. Identification of Differential Lipid Molecules and Functional Enrichment Analysis

Quality control (QC) samples were analyzed throughout the sequence to ensure data stability. Features with a relative standard deviation (RSD%) > 30% in QC samples were excluded. The acquired MS/MS data were processed using Analyst v1.6.3 and MultiQuant software v3.0.3 for peak integration and quantification. The differential lipid molecules (DLMs) were defined using orthogonal partial least squares–discriminate analysis (OPLS-DA) in MetaboAnalyst v4.0, with thresholds of variable importance in projection (VIP) ≥ 1.0, *p*-value < 0.05 and |log_2_(fold change)| ≥ 1. Lipid subclass dynamics were analyzed by ANOVA with polynomial contrasts for trend analysis. Fatty acid composition was characterized by carbon chain length (Σ carbon number) and saturation level (Σ double bonds) for major subclasses. Enriched metabolic pathways for DLMs were identified using the KEGG database (*p*-value < 0.05).

### 2.8. Statistical Analysis

All data were subjected to comprehensive statistical analysis using validated methodologies. Significant differences among different overfeeding stages (D0, D16, D25) were determined by one-way analysis of variance (ANOVA) followed by Tukey’s honestly significant difference (HSD) post hoc test for multiple comparisons, with significance thresholds set at *p*-value < 0.05 and *p*-value < 0.01. Homogeneity of variance was verified using Levene’s test. Results are denoted by letter superscripts where identical letters indicate non-significant differences. GraphPad Prism 8.0 software (GraphPad Software, Inc., San Diego, CA, USA) was used to visualize the statistical results. To ensure statistical power, a post hoc power analysis was performed using G*Power v3.1 software [39] for all key datasets (phenotypic parameters, n = 6; transcriptomics and lipidomics, n = 3). The results confirmed that the sample sizes provided sufficient statistical reliability.

## 3. Results

### 3.1. Overfeeding Dramatically Changed the Global Appearance, Hepatic Histology and Chemical Composition of Goose Livers

In this study, overfeeding markedly altered the appearance of the geese liver, resulting in a color change of the liver from reddish-brown to yellow and the size of the liver increased significantly (Figure 1A–C). In the D0 group, HE staining of the liver revealed that hepatocytes were homogeneous in size, with nuclei centrally located. Over time, following overfeeding, the size of the hepatocytes continued to increase, with nuclei displaced toward the periphery. Significant lipid accumulation was observed within the hepatocytes, with large lipid droplets evident in the D16 and D25 groups (Figure 1D–F). As shown in Table 1, the body weight, liver weight and liver index increased significantly after overfeeding (*p*-value < 0.01). And moisture, crude ash (CA), and crude protein (CP) contents declined continuously (*p*-value < 0.01) after overfeeding. In contrast, crude fat (CF) content increased substantially, with an approximately eightfold rise in the D25 group compared to the D0 group (*p*-value < 0.01).

### 3.2. Analysis of Gene Expression Levels in Goose Livers

To explore regulatory genes influencing goose fatty liver formation, we performed transcriptome sequencing on liver samples from geese at the D0, D16, and D25 groups. Principal Component Analysis (PCA) (Figure 2A) showed that gene expression profiles of goose livers distinctly separated across the overfeeding stages. The clustering of gene expression heatmap (Figure 2B) further revealed clear differences in gene expression patterns among the three groups. Collectively, the most pronounced differences were observed between the D0 group and D16 group.

### 3.3. Identifying Differential Expression Genes and Key Pathways at Different Overfeeding Stages

Comparing D16 group and D0 group, 497 DEGs were detected: 274 upregulated and 223 downregulated (Figure 3A). These genes are involved in key processes of fat metabolism, including fatty acid synthesis (*ACLY*), fatty acid oxidation (*PPARGC1A*), hepatic lipid metabolism (*ACSBG2*, *CPT1B*), fatty acid uptake (*FABP1*), profibrogenesis (*ADAMTSL2*, *AEBP1*, *COL6A3*), and triacylglycerol synthesis (*DGAT2*). Comparing D25 group and D16 group, 303 DEGs were detected: 191 upregulated and 112 downregulated (Figure 3B). Genes involved in this process included: lipoprotein lipase inhibitor (*ANGPTL4*), fatty acid oxidation (*CPT1B*, *PPARGC1A*), triacylglycerol synthesis (*DGAT2*), fatty acid uptake (*FABP1*), and lipid synthesis (*NR1D1*). The higher number of DEGs detected between the D0 group and D16 groups indicates significant changes in goose liver gene expression during this period, consistent with the pattern of fat deposition in goose livers.

The 450 DEGs were unique to the D16 vs. D0 comparison, 256 were unique to the D25 vs. D16 comparison, and 47 DEGs were shared between two comparisons (Figure 3C). These shared DEGs included genes such as the fatty acid-binding protein gene *FABP1*, the key triacylglycerol synthesis gene *DGAT2*, and the fatty acid oxidation regulator *PPARGC1A*. Notably, *EFHD1*, which has been reported to be regulated by *FOXO1*, was among them; the absence of its eQTL can lead to disordered hepatic lipid metabolism. This result suggests these shared genes may play key regulatory roles in the fat deposition process of goose fatty livers [40].

KEGG pathway enrichment analysis was performed on DEGs from two comparisons, with the top 20 pathways ranked by *p*-value displayed (Figure 3D,E. DEGs from the D16 vs. D0 comparison (249 genes) were significantly enriched in pathways including PPAR signaling pathway, metabolic pathways, carbon metabolism, pentose phosphate pathway, biosynthesis of amino acids, cell cycle, FoxO signaling pathway, fatty acid metabolism, butanoate metabolism, glycerolipid metabolism, and biosynthesis of unsaturated fatty acids (Figure 3D; Appendix A). DEGs from the D25 vs. D16 comparison (168 genes) were significantly enriched in pathways such as metabolic pathways, FoxO signaling pathway, MAPK signaling pathway, glycerophospholipid metabolism, adipocytokine signaling pathway, and PPAR signaling pathway. Most metabolic pathways were closely related to lipid metabolism (Figure 3E; Appendix A).

### 3.4. Analysis of Lipid Composition in Goose Livers

We assessed the lipid dynamic changes of goose livers across the overfeeding stages using targeted lipidomic approach. The PCA results revealed three distinct clusters between D0, D16 and D25 groups (Figure 4A). The total of 1274 lipid molecules were identified, including 671 in positive mode and 603 in negative mode (Appendix A). Based on their structures, the lipid molecules in both modes were classified into six categories: glycerolipids (GL), glycerophospholipids (GP), fatty acyls (FA), sphingolipids (SP), sterol lipids (ST), and prenol lipids (PR). The number of lipid molecules varied significantly among categories, with GP and GL representing the major lipid classes, together accounting for over 85.4% of the total lipid molecules (Figure 4B,C). To differentiate the lipid composition of goose liver across different overfeeding stages, the lipidomic data was analyzed using multivariate statistics. The cluster dendrogram of lipid content revealed clear discrimination among the overfeeding periods (Figure 4D), indicating that overfeeding dramatically altered the lipid composition of goose livers.

For functional analysis, the lipid molecules were further divided into 44 subclasses (Figure 5). In this study, the number of lipid subclasses assigned into the categories of GL, GP, FA, SP, ST, and PR were 5, 19, 3, 12, 4, and 1, respectively. Among the lipid subclasses, triacylglycerols (TG), phosphatidylethanolamines (PE), and phosphatidylcholine (PC) were the most abundant, both in terms of the number of lipid molecules and the average relative intensity. The results showed a continuous increase (*p*-value < 0.05) in the relative content of TG, diglycerides (DG), Diglyceride-O-alkyl (DG-O), non-hydroxy sphingosine hexosylceramides (HexCer-NS), alpha-hydroxy sphingosine ceramide (Cer-AS), non-hydroxy phytosphingosine ceramide (Cer-NP), non-hydroxy dihydrosphingosine ceramide (Cer-NDS), and alpha-hydroxy phytosphingosine hexosylceramide (HexCer-AP). Conversely, the relative content of lysophosphatidic acid (LPA), lysophosphatidylserine (LPS), and bile acids (BA) decreased continuously (*p*-value < 0.05). Additionally, we observed a tendency of decrease for the relative content of PE, PC, phosphatidylserine (PS), phosphatidylglycerol (PG), phosphatidic acid (PA), lysophosphatidic acid (LNA) and eicosanoids (*p*-value > 0.05). In summary, the increased lipid subclasses were primarily GL and SP, while the decreased subclasses were mainly GP and ST.

### 3.5. Dynamic Changes of Fatty Acid Composition in Each Lipid Subclass with Respect to the Carbon Chain Length and Saturation

The fatty acid compositions of major lipid subclasses are presented in Figure 6A. Our results indicated that the fatty acid profiles varied among the lipid subclasses regarding carbon chain saturation. More than 95% of TG, PC, and PE molecules contained more than one double bond, indicating a high degree of unsaturated fatty acids in these subclasses. In contrast, PG and PS were rich in saturated fatty acids, with over 40% of molecules containing no double bonds. Across different overfeeding stages, the saturation levels of fatty acids in PC, PE, PG, and PS remained similar. However, the saturation of DG molecules was affected by overfeeding. The percentage of DG molecules with zero to four double bonds exhibited a continuous decrease, while those with one to two double bonds showed a tendency of continuous increase. Notably, the abundance of TG molecules containing more than eight double bonds increased dramatically from D0 to D16 (*p*-value < 0.01) (Figure 6B).

Regarding carbon chain length, TG molecules with a total of 52 and 54 carbons, DG and PC with 34 and 36 carbons, PE and PG with 36 and 38 carbons were the most abundant, respectively (Figure 7A,B). Notably, PS molecules containing 37 and 39 carbons were also observed to be particularly abundant. Furthermore, across the different overfeeding stages, the percentage of PS molecules with 37 and 39 carbons increased, while the percentage of PS molecules with 36 and 40 carbons decreased. In contrast, DG molecules containing 34 and 36 carbons exhibited an increasing trend, whereas those with 32 and 38 carbons showed a decreasing trend.

### 3.6. Identifying Differential Lipid Molecules and Key Pathways at Different Overfeeding Stages

DLMs in goose livers were screened based on the OPLS-DA model with thresholds of *p*-value < 0.05, VIP ≥ 1.0, and |log_2_(fold change)| ≥ 1. Comparing the D0 group, 256 DLMs were upregulated in D16 group, primarily consisting of TG (76.17%) and DG (9.38%). Conversely. 112 DLMs were downregulated in D16 group, with the majority being PE (11.60%), PC (11.60%), PE-O (17.86%) and PS (10.71%) (Figure 8A). Comparing the D16 group, 127 DLMs were upregulated in D25 group, with TG and DG accounting for 46.46% and 30.71%, respectively. The D25 group exhibited 45 downregulated DLMs, predominantly PC (31.11%) and LPC (11.11%) (Figure 8B). A total of 368 and 172 DLMs were identified in the comparisons of D16 vs. D0 and D25 vs. D16 comparisons, respectively. Additionally, 90 common DLMs were observed across both comparisons (Figure 8C).

To investigate the metabolic pathways involved in fatty liver formation, the DLMs were analyzed using KEGG pathway. For the DLMs in the comparison of D16 vs. D0, the enriched KEGG pathways primarily include insulin secretion, regulation of lipolysis in adipocytes, lipid and atherosclerosis, cholesterol metabolism, vitamin digestion and absorption, fat digestion and absorption, thermogenesis, and glycerophospholipid metabolism (Figure 8D; Appendix A). In D25 vs. D16 comparison, the enriched KEGG pathways include long-term depression, inositol phosphate metabolism, teichoic acid biosynthesis, glycerolipid metabolism, phosphatidylinositol signaling system, fat digestion and absorption, regulation of lipolysis in adipocytes, thermogenesis, lipid and atherosclerosis, vitamin digestion and absorption, cholesterol metabolism, and insulin resistance (Figure 8E; Appendix A).

### 3.7. Integrated Lipidomic and Transcriptomic Analysis in Goose Fatty Livers

To further explore the regulatory mechanisms underlying goose fatty liver formation after overfeeding, we conducted an integrated analysis of enriched KEGG pathways from DLMs and DEGs. Our results showed that the KEGG pathways significantly enriched by both DLMs and DEGs from the D16 vs. D0 comparison converged on the Glycerolipid metabolism pathway (Figure 9A). KEGG pathways significantly enriched by both DLMs and DEGs from the D25 vs. D16 comparison converged on glycerolipid metabolism, adipocytokine signaling pathway, MAPK signaling pathway, and ErbB signaling pathway, suggesting their potential critical roles in fatty liver development (Figure 9B). Based on these four key metabolic pathways, we constructed a regulatory network for goose fatty liver formation during overfeeding (Figure 9C, Table 2). This network ultimately drives TG synthesis and deposition, representing a crucial factor in goose fatty liver development induced by excessive force-feeding.

The regulatory network (Figure 9C) integrates our core findings, proposing a mechanistic model for overfeeding-induced steatosis. This model positions glycerolipid metabolism as the central pathway responsible for TG and DG accumulation. Key genes within this pathway function coordinately: LIPG hydrolyzes phospholipids to release fatty acids, increasing substrate availability; *DGAT2* is upregulated, catalyzing the conversion of DG to TG and driving lipid droplet expansion; and *LPIN1* support DG generation. This metabolic core is regulated by three signaling pathways. The Adipocytokine pathway, involving *SLC2A1* and *PPARGC1A*, adjusts hepatic energy status. The MAPK and ErbB pathways are activated, evidenced by induced expression of *AREG*, *DUSP1*, and *DUSP10*, mediating cellular stress responses and potentially reinforcing lipogenic gene expression. Critically, diglyceride (DG) serves as a central node, functionally interconnecting these pathways. It acts not only as the direct precursor for TG synthesis but also as a potential lipid second messenger, potentially influencing insulin and MAPK signaling and creating a feed-forward loop that exacerbates lipid deposition. Collectively, overfeeding signals are transduced through the Adipocytokine, MAPK, and ErbB pathways, converging to hyperactivate glycerolipid metabolism and culminate in massive TG accumulation.

## 4. Discussion

Goose foie gras is a premium functional food with significant economic value and broad global market prospects. Understanding its formation mechanism is crucial to meet increasing consumer demand for enhanced nutritional quality. This study employed integrated lipidomic and transcriptomic analyses to systematically elucidate dynamic lipid changes and underlying transcriptional regulatory mechanisms in goose liver during overfeeding. These findings provide novel insights into the molecular basis of goose fatty liver formation.

Lipids are fundamental structural components of animal organisms, playing pivotal roles in energy supply and essential structural/metabolic functions [41,42]. Dysregulation of lipid metabolism is a key driver of fatty liver development [43]. In this study, GP, GL, and SP were the primary lipid classes in goose fatty livers. Among these, TG, PE, and PC were the most stable subclasses, consistent with previous reports [3,44,45]. Goose liver is renowned for its high unsaturated fatty acid content [46]. Lipidomic analysis revealed distinct fatty acid profiles across subclasses: TG, PC, and PE were enriched in UFAs, while DG, PG, and PS contained predominantly saturated fatty acids. Overfeeding significantly altered fatty acid saturation and carbon chain length within TG and DG molecules [3,11], indicating hepatic fatty acid metabolic pathway remodeling. This aligns strongly with transcriptomic enrichment of KEGG pathways like “fatty acid metabolism” and “biosynthesis of unsaturated fatty acids”, suggesting that expression changes in key regulatory genes (the fatty acid synthase: *FASN*, *ACACA*; the desaturases: *FADS1*, *FADS2*) likely drove these lipid compositional shifts. Notably, although glycerophospholipids (as major membrane components) are thought to promote hepatocyte enlargement during overfeeding [47,48], we observed a consistent decline in the relative content of LPA, PE, PC, and PS [3,49]. The mechanism underlying this discrepancy warrants further investigation.

TG and DG serve as primary storage lipids [50], and their abnormal accumulation is a hallmark of goose fatty liver formation. Overfeeding significantly altered the hepatic lipid profile. Lipidomic data showed a continuous, substantial increase in TG/DG levels across all overfeeding stages [3,22,44,50], leading to massive lipid droplet accumulation in hepatocytes [11]. This lipid accumulation phenotype correlated highly with transcriptomic findings. Previous studies indicate that lipoprotein lipase (*LPL*) expression changes significantly regulate TG hydrolysis and uptake [51]. Upregulated diacylglycerol O-acyltransferase 2 (*DGAT2*) directly promotes DG-to-TG conversion, driving TG deposition in lipid droplets and acting as the direct executor of goose fatty liver formation [52]. As a therapeutic target, *DGAT2* has demonstrated significant efficacy in clinical interventions for NAFLD [53]. Endothelial lipase (*LIPG*) hydrolyzes phospholipids (e.g., PC) in high-density lipoprotein, releasing free fatty acids that increase DG synthesis substrate availability [53,54,55], thereby facilitating *DGAT2*-catalyzed TG synthesis. Under insulin resistance, this *LIPG*-mediated free fatty acids release may exacerbate hepatic TG deposition and link to NAFLD pathogenesis [56,57]. Genes involved in de novo fatty acid synthesis, such as ATP citrate lyase (*ACLY*), influence hepatic fatty acid synthesis capacity, affecting TG synthesis substrate supply [58]. Collectively, these key DEGs (*LPL*, *ACLY*, *DGAT2*, *LIPG*) provide a crucial transcriptional explanation for abnormal TG/DG accumulation.

Glycerolipid metabolism plays a central role in goose fatty liver formation. Integrated lipidomic-transcriptomic analysis identified four key KEGG pathways: glycerolipid metabolism, adipocytokine signaling, MAPK signaling, and ErbB signaling. Glycerolipid metabolism was significantly co-enriched in both comparison groups (D16 vs. D0 and D25 vs. D16). This pathway directly regulates TG and DG synthesis, degradation, and interconversion. The persistent rise in TG/DG levels observed lipidomically represents its direct functional output. Concurrently, transcriptomics revealed significant differential expressions of key pathway genes (the glycerol phosphorylase: *LPIN1*, *LPIN2*; the acyltransferase: *DGAT2*). This strongly establishes glycerolipid metabolism as the core hub linking transcriptional regulation (gene expression changes) to phenotypic output (TG/DG accumulation), critically functioning throughout goose fatty liver development. The adipocytokine signaling pathway mediates adipose tissue-derived signals (such as non-esterified fatty acid supply and adipokines), which affect hepatic lipid metabolism and insulin sensitivity [59,60,61]. The MAPK and ErbB signaling pathways likely regulate hepatocyte responses to metabolic stress, inflammation, and proliferation/hypertrophy processes [47,48], along with crosstalk with PI3K-Akt signaling pathway, collectively maintaining the high lipid accumulation state.

Lipid accumulation and insulin resistance form a positive feedback loop. Lipidomic functional analysis indicated significant enrichment of the “insulin secretion” pathway during the D16 vs. D0 comparison. Elevated TG levels may induce insulin resistance [62], consistent with NAFLD pathogenesis [63]. Transcriptomic data simultaneously showed FoxO signaling pathway enrichment. FoxO, a key downstream effector of insulin signaling, regulates glucose/lipid metabolism, oxidative stress, and apoptosis [64]. Its activation status, modulated by insulin levels, may reflect or mediate hepatic insulin sensitivity changes. Furthermore, insulin resistance can exacerbate metabolic dysregulation through feedback mechanisms.

## 5. Conclusions

We successfully established a goose fatty liver model, observing significant increases in liver index after overfeeding. Changes in global appearance, hepatic histology and chemical composition confirmed the typical characteristics of goose fatty liver development. Furthermore, we identified significant temporal synergy between the evolution of the hepatic lipid profiles (notably the explosive accumulation of triacylglycerols and diglycerides) and the differential gene expression regulatory network. Glycerolipid metabolism served as the central hub throughout the overfeeding process, while adipocytokine signaling, MAPK, and ErbB pathways played key synergistic regulatory roles, particularly during the mid-to-late stages of lipid deposition, collectively driving the physiological formation of goose fatty liver. Through integrated lipidomic and transcriptomic analyses, this study systematically revealed key regulatory lipid molecules and genes involved in this process. These findings provide novel insights into the molecular mechanisms underlying fatty liver development in geese, establishing a theoretical foundation for further research. The established goose model also serves as a valuable in vivo system for studying human NAFLD. Furthermore, this foundational knowledge enables the development of in vitro cell-culture strategies to produce cultured alternatives to foie gras, thereby eliminating the need for traditional force-feeding practices and enhancing animal welfare standards.

## Figures and Tables

**Figure 1 biology-14-01617-f001:**
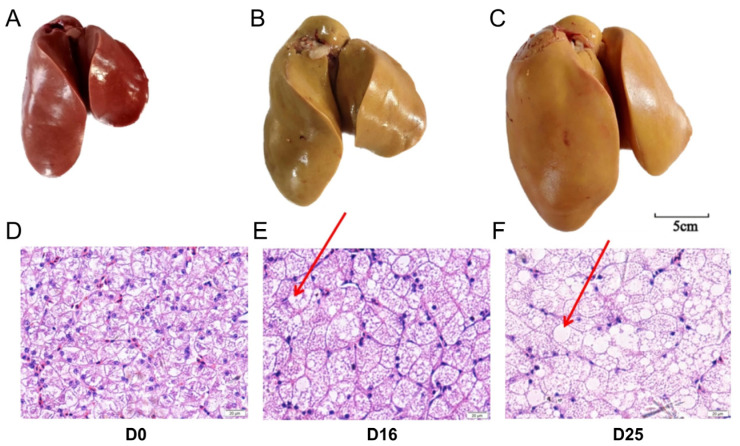
The global appearance and hepatic histology of goose livers at different overfeeding stages (D0, D16 and D25). (**A**–**C**) The appearance of goose livers from D0 (**A**), D16 (**B**) and D25 (**C**) groups. (**D**–**F**) The HE Stained liver tissue sections (40×) of goose livers from D0 (**D**), D16 (**E**) and D25 (**F**) groups; red arrows indicate large lipid droplets.

**Figure 2 biology-14-01617-f002:**
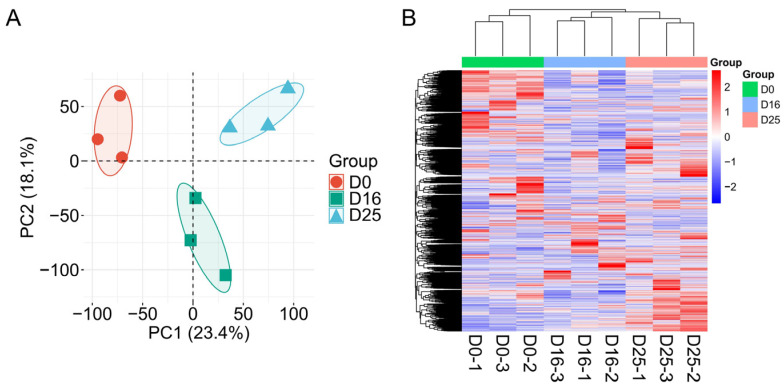
Gene expression of goose livers at different overfeeding stages (D0, D16 and D25). (**A**) Principal component analysis (PCA) analysis of gene expression profiles across different overfeeding stages. (**B**) Gene expression heatmap across different overfeeding stages.

**Figure 3 biology-14-01617-f003:**
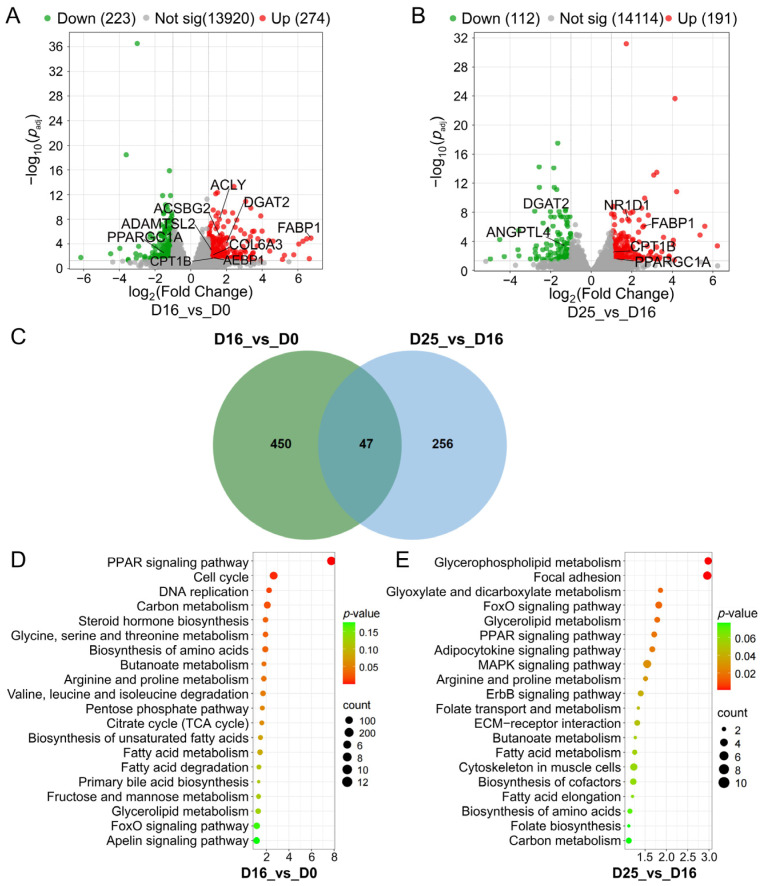
Identification and KEGG analysis of differentially expressed genes (DEGs). (**A**) Volcano plot of DEGs in D16 vs. D0 comparison. (**B**) Volcano plot of DEGs in D25 vs. D16 comparison. (**C**) Venn diagram of DEGs between D16 vs. D0 and D25 vs. D16 comparisons. (**D**) KEGG analysis of DEGs in D16 vs. D0 comparison. (**E**) KEGG analysis of DEGs in D25 vs. D16 comparison.

**Figure 4 biology-14-01617-f004:**
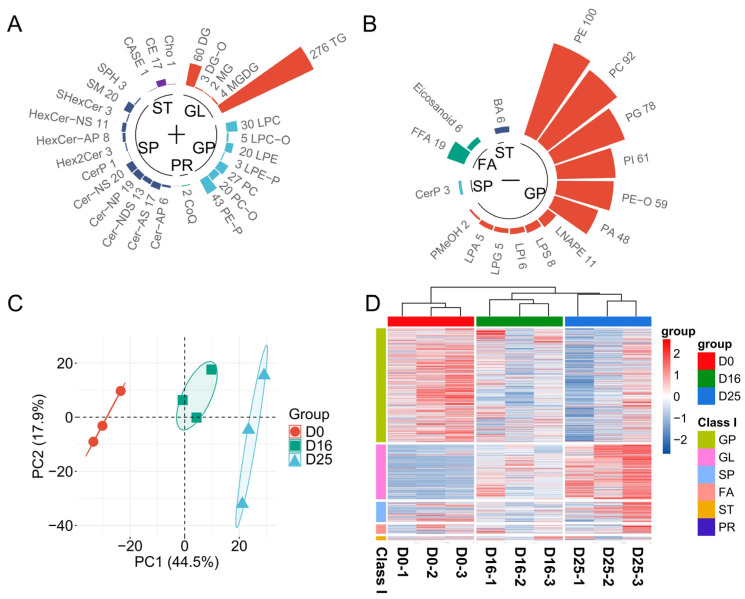
Overview of lipid profiles at goose livers at different overfeeding stages (D0, D16 and D25). (**A**) PCA analysis of lipid profiles at different overfeeding stages. (**B**,**C**) Number of lipid molecules identified in positive and negative mode. The different color represents lipid classes. The polar bar length indicates the number of lipid molecules identified in each subclass. (**D**) Heatmap of goose lipid profiles at the overfeeding stages. Each row corresponds to a lipid molecule level.

**Figure 5 biology-14-01617-f005:**
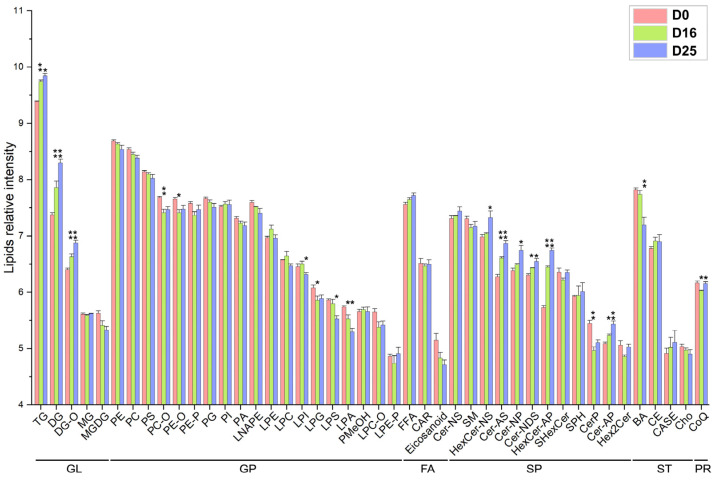
The average relative content of lipid molecules in each subclass at different overfeeding stages (D0, D16 and D25). * *p*-value < 0.05; ** *p*-value < 0.01.

**Figure 6 biology-14-01617-f006:**
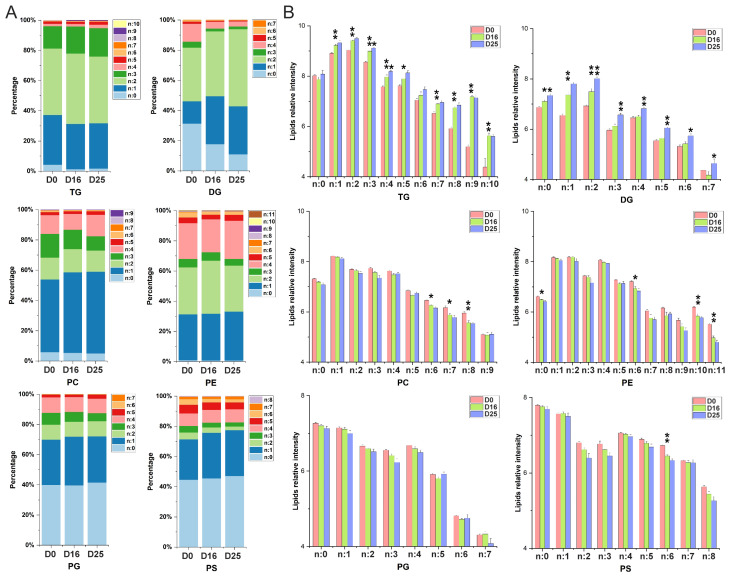
Composition of major lipid subclasses by carbon chain saturation. (**A**) The percentage of lipid molecules in major lipid subclasses with respect to the summed carbon chain saturation. (**B**) The relative content of lipid molecules in major lipid subclasses with respect to the summed carbon chain saturation. * *p*-value < 0.05; ** *p*-value < 0.01.

**Figure 7 biology-14-01617-f007:**
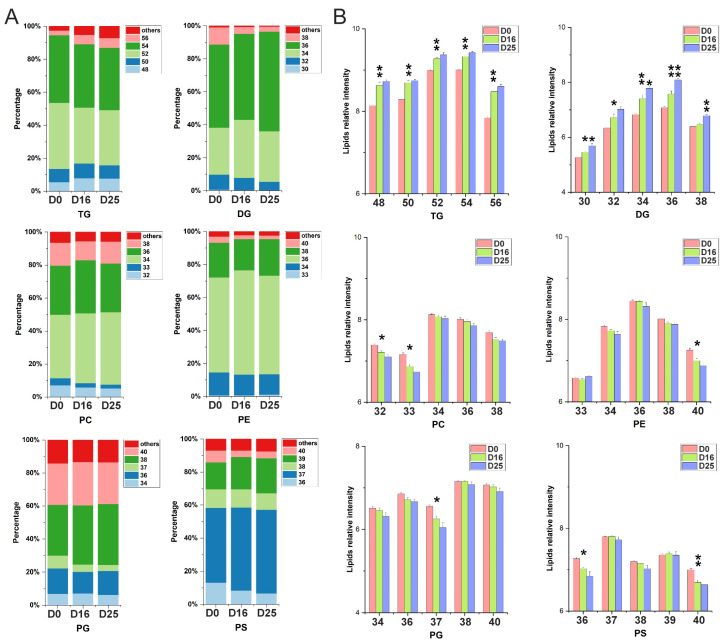
Composition of major lipid subclasses by carbon chain length. (**A**) The percentage of lipid molecules in major lipid subclasses with respect to the summed carbon chain length. (**B**) The relative content of lipid molecules in major lipid subclasses with respect to the summed carbon chain length. * *p*-value < 0.05; ** *p*-value < 0.01.

**Figure 8 biology-14-01617-f008:**
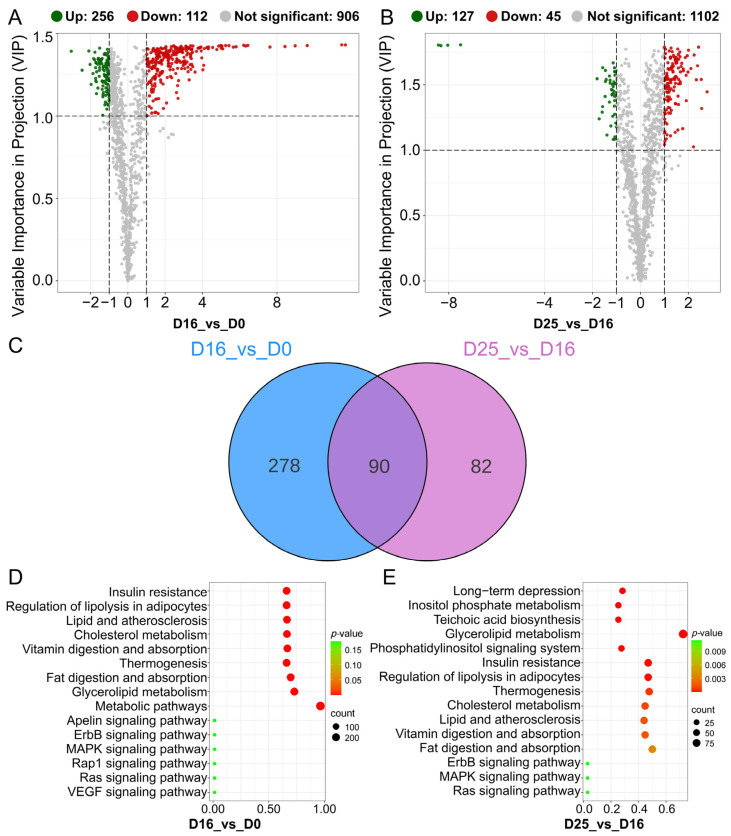
Identification and KEGG analysis of differential lipid molecules (DLMs) at different overfeeding stages (D0, D16 and D25). (**A**) Volcano plot of DLMs in D16 vs. D0 comparison. (**B**) Volcano plot of DLMs in D25 vs. D16 comparison. (**C**) Venn diagram of DLMs between D16 vs. D0 and D25 vs. D16 comparisons. (**D**) The top 15 enriched KEGG pathways of DLMs in D16 vs. D0 comparison. (**E**) The top 15 enriched KEGG pathways of DLMs in D25 vs. D16 comparison.

**Figure 9 biology-14-01617-f009:**
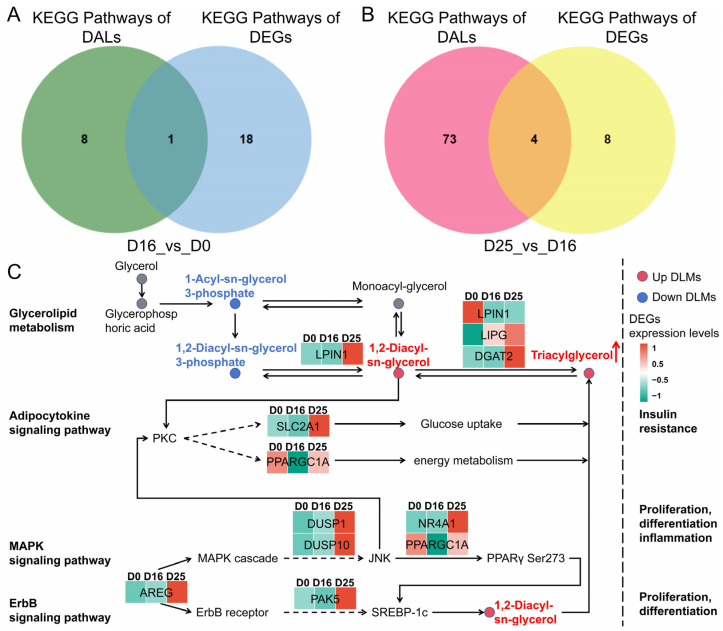
Integrated analysis of transcriptomics and lipidomics in overfed goose fat livers. (**A**) Venn diagram showing KEGG pathways co-enriched by DALs and DEGs (D16 vs. D0). (**B**) Venn diagram showing KEGG pathways co-enriched by DALs and DEGs (D25 vs. D16). (**C**) The regulated network involved glycerolipid metabolism, adipocytokine signaling pathway, MAPK signaling pathway and ErbB signaling pathway; the red arrow indicates triacylglycerol accumulation.

**Table 1 biology-14-01617-t001:** The weight and chemical composition of goose livers at different overfeeding stages (D0, D16 and D25).

Item	D0	D16	D25	SEM	*p*-Value
Body weight (kg)	4.09 ^b^	5.83 ^a^	5.51 ^a^	0.289	<0.01
Liver weight (g)	183.83 ^c^	353.60 ^b^	502.45 ^a^	47.784	<0.01
Liver index (%)	3.87 ^c^	6.08 ^b^	9.36 ^a^	0.873	<0.01
Moisture (%)	64.58 ^a^	52.38 ^b^	39.85 ^c^	3.581	<0.01
Crude ash (%)	1.28 ^a^	0.81 ^b^	0.52 ^c^	0.114	<0.01
Crude fat (%)	5.52 ^c^	25.21 ^b^	43.85 ^a^	5.537	<0.01
Crude protein (%)	15.96 ^a^	11.84 ^b^	9.35 ^b^	0.972	<0.01

Note: Liver index = (Liver weight/Body weight) × 100%. Different lowercase letters in the peer corner indicate significant differences, while the same letters indicate insignificant differences. n = 6 per group.

**Table 2 biology-14-01617-t002:** Key signaling pathways and associated DEGs and DLMs in the formation of goose fatty liver.

Comparison	KEG ID	KEGG Term	KEGG Term	DLMs
D16 vs. D0	ko00561	ko00561	*ALDH7A1*, *DGAT2*, *LIPG*, *LPL ALDH7A1*, *DGAT2*, *LIPG*, *LPL*	1,2-Diacyl-sn-glycerol 3-phosphate, 1,2-Diacyl-sn-glycerol (Diglyceride), Triacylglycerol
D25 vs. D16	ko00561	Glycerolipid metabolism	*DGAT2*, *DGKI*, *LPIN1*, *LPIN2*	1-Acyl-sn-glycerol3-phosphate, 1,2-Diacyl-sn-glycerol (Diglyceride), Triacylglycerol
	ko04920	Adipocytokine signaling pathway	*ACSBG2*, *NFKBIA*, *PPARGC1A*, *SLC2A1*	Diglyceride
	ko04010	MAPK signaling pathway	*AREG*, *DUSP1*, *DUSP10*, *DUSP16*, *EPHA2*, *FOS*, *IGF1*, *NR4A1*, *PGF*	Diglyceride
	ko04012	ErbB signaling pathway	*AREG*, *CDKN1A*, *HBEGF*, *PAK5*	Diglyceride

## Data Availability

The raw reads obtained from the RNA sequencing are publicly available at the Genome Sequence Archive under the accession number: CRA028714.

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
