# Peer review of "Lipidomic and Transcriptomic Reveals Variations in Lipid Deposition During Goose Fatty Liver Formation"

_biology, 2025, doi:10.3390/biology14111617_

Round 1
Reviewer 1 Report
Comments and Suggestions for Authors
Dear Authors,
I find Your work as very interesting. The paper presents a well-designed and detailed study combining lipidomic and transcriptomic analyses to explain how fatty liver develops in Landes geese. The introduction is clear, provides good background information, and includes up-to-date references. The experimental design is appropriate, and the results are clearly described and logically discussed.
Nonetheless, some small improvements are recommended. Firs of all, the methods section is very long. Technical details could be shortened or moved to the supplementary materials. Secondly, I think that You need to improve readability of some figures and tables-for example, Figures 6 and 7 are very complex and difficult to interpret without magnification. Lebels and numbering in captions should be more legible. The last but not the least, the figure showing the integrated pathways should explain more clearly how genes and metabolites are connected. It is not easy to understend it in this form of presentation.
Overall, this is an interesting and valuable study that makes a real contribution to understanding lipid metabolism in geese and its relation to fatty liver disease. In my opinion, Your paper can be accepted after minor revisions.
Kind regards :)
Reviewer 2 Report
Comments and Suggestions for Authors
Lines 55-56: This statement requires reference.
Lines 97-98: The authors mention that geese were provided with sufficient living space in cages (3 birds per cage, ≥0.25 m²/bird) with constant water access. Could the authors please clarify whether this space allowance is based on a specific guideline or reference supporting that it is adequate for geese?
Line 101: The composition of the feed used in the study is missing. Including details about the diet composition would help readers better understand and interpret the results.
Lines 103-105: The description of the force-feeding procedure is clear; but the manuscript does not specify how many times per day the geese were fed or at what times the feedings occurred. Pleae provide these information.
Lines 113: Delete %
M&M Section: I would recommend you to make a power analysis as the number of geese seems to be low.
Line 135: A reference should be provided for the transcriptomic analysis section to support the methodology
Figure 4E: As this figure is highly important, I recommend enlarging it so that readers can better understand the results. It is currently too small.,
Figure 4E,5E-D: In the multiple comparison test, the stars are shown inconsistently. For example, some columns have one, two, or three stars, while others have none. It is unclear which groups are significantly different. Please clarify this, or consider using letters (a, b, c) instead of stars to show the differences more clearly.
